# Host Calcium Channels and Pumps in Viral Infections

**DOI:** 10.3390/cells9010094

**Published:** 2019-12-30

**Authors:** Xingjuan Chen, Ruiyuan Cao, Wu Zhong

**Affiliations:** 1Institute of Medical Research, Northwestern Polytechnical University, Xi’an 710072, China; xjchen@nwpu.edu.cn; 2National Engineering Research Center for the Emergency Drug, Beijing Institute of Pharmacology and Toxicology, Beijing 100850, China

**Keywords:** virus, calcium channels, calcium pumps, virus–host interaction, antiviral

## Abstract

Ca^2+^ is essential for virus entry, viral gene replication, virion maturation, and release. The alteration of host cells Ca^2+^ homeostasis is one of the strategies that viruses use to modulate host cells signal transduction mechanisms in their favor. Host calcium-permeable channels and pumps (including voltage-gated calcium channels, store-operated channels, receptor-operated channels, transient receptor potential ion channels, and Ca^2+^-ATPase) mediate Ca^2+^ across the plasma membrane or subcellular organelles, modulating intracellular free Ca^2+^. Therefore, these Ca^2+^ channels or pumps present important aspects of viral pathogenesis and virus–host interaction. It has been reported that viruses hijack host calcium channels or pumps, disturbing the cellular homeostatic balance of Ca^2+^. Such a disturbance benefits virus lifecycles while inducing host cells’ morbidity. Evidence has emerged that pharmacologically targeting the calcium channel or calcium release from the endoplasmic reticulum (ER) can obstruct virus lifecycles. Impeding virus-induced abnormal intracellular Ca^2+^ homeostasis is becoming a useful strategy in the development of potent antiviral drugs. In this present review, the recent identified cellular calcium channels and pumps as targets for virus attack are emphasized.

## 1. Introduction

Viruses exploit the environment of host cells to replicate, thereby inducing host cells’ dysfunction. Virus–host interaction is the foundation of pathogenesis and closely associated with disease severity and incidence. The prevention and therapy of virus infections are often confounded by the high mutation rates that facilitate the viral evasion of antiviral strategies that target virally encoded proteins. Modulations of the intracellular environment have become an important strategy in antiviral drug discovery and development. In mammalian cells, Ca^2+^, as an important second messenger, mediates the sensor input and responses output for almost all known cellular progress, such as stress responses, synaptic plasticity, immunodefenses, protein transport, and endosome formation [1,2]. It has been demonstrated that the host cell dysfunction following infection with a virus is accompanied by abnormal intracellular Ca^2+^ concentration [3]. A virus can hijack the host intracellular Ca^2+^ system to achieve successful replication via multiple routes; for instance, viral proteins directly bind to Ca^2+^ or disturb the membrane permeability for Ca^2+^ by manipulating Ca^2+^ apparatus. 

The host cell plasma membrane is the first barrier against the invasion of viruses. Various Ca^2+^ channels and pumps are distributed on the cell plasma membrane. Therefore, these membrane proteins become the direct target of virus infection. Interaction between viruses and these membrane proteins is the foremost approach of viruses perturbing the host cell calcium signal system. This interaction may inhibit or stimulate calcium influx and modulate free cytosolic Ca^2+^ concentrations. After entry into the host cell, viruses stimulate or inhibit the calcium release from internal stores via an effect on calcium-permeable channels, transporters, and exchangers on organellar membranes. Then, the change in cytosolic calcium concentration may trigger further distortion of the host cell system, which benefits virus survival and replication.

This review concentrates on host cell membranes’ calcium channels and pumps in viral infection. Blockers for these membrane proteins or preventing viruses from grabbing these host calcium-signaling components may lower the probability of virus stability, replication, and release, as well as infection-related host–cell apoptosis and reactive oxygen species production, neurotoxicity, and enterotoxin, making these membrane proteins potential targets for antiviral drugs.

## 2. Calcium Channels and Pumps in Host Ca^2+^ Homeostasis

Cellular Ca^2+^ is from two major sources: the internal Ca^2+^ store (mainly endoplasmic reticulum (ER) or sarcoplasmic reticulum (SR)) and the extracellular medium. Calcium channels on cell plasma membrane mediate the entry of Ca^2+^ from the extracellular medium. These channels are activated by specific stimuli, such as voltage-gated calcium channels (VGCCs), which are stimulated by membrane depolarization, specific receptor-operated channels (ROC), which are stimulated by external agonists, or intracellular messengers and store-operated calcium channel (SOC), which are stimulated by the depletion of internal Ca^2+^ stores. The IP_3_ receptor (IP_3_R) and the ryanodine receptors (RyR) are the main players in mediating the release of Ca^2+^ from the internal stores. Inositol-1,4,5-triphosphate (IP_3_) activates IP_3_R, triggers Ca^2+^ release from stores, and further increases IP_3_R’s sensitivity to Ca^2+^. Calcium pumps (the plasma membrane Ca^2+^-ATPase (PMCA), sarcoplasmic/endoplasmic reticulum Ca^2+^-ATPase (SERCA)) and the Na^+^/Ca^2+^ exchanger (NCX) are responsible for transporting Ca^2+^ from the cytosol to external medium or into cellular calcium stores (Figure 1). The normal function of these calcium channels and pump is important for cells to maintain intracellular Ca^2+^ homeostasis. 

These channels and pumps are activated in a flexible and precise manner to generate specific Ca^2+^ signaling, satisfying various spatiotemporal requirements. During the viral infections, host cells modulate these calcium-signaling components in response to the infection. On the other hand, viruses utilize these components to create a cellular environment that benefits their own lifecycles [4]. Viruses induce elevated cytosolic calcium concentration to activate Ca^2+^ dependent/sensitive enzymes and transcriptional factors to promote virus replication. Mitochondria could take the Ca^2+^ to generate more energy to support continuous virus replication. Moreover, regulating the calcium concentration in ER or Golgi may inhibit host proteins trafficking and promote virus protein maturation. The inhibition of host proteins frustrates host antiviral immune responses, while the promotion of virus protein maturation benefits virus propagation (Figure 1). The journey begins when a virus encounters the host cell and attaches to the cell surface. The virus particle penetrates the cytoplasm via direct membrane fusion or receptor-mediated endocytosis followed by exposing its genome to cellular machinery for replication. When the viral proteins and viral genomes are accumulated, they are assembled to form a progeny virion particle and then released [5]. During the viral lifecycles, they utilize various calcium channels and pumps to obviate the cell membrane barriers, enter the host cell, complete replication, acquire infection ability, and release.

## 3. Viruses Control Host Voltage-Gated Calcium Channels (VGCCs) and Two-Pore Channels (TPCs)

VGCCs are widely found in the membrane of excitable cells [6] and one of the most well studied viral targets because of the availability of specific channel blockers. There are several different types of VGCCs: L-type (Long-conducting) channels are mostly found in skeletal and smooth myocytes, bone (osteoblasts), and ventricular myocytes; N-type (Non-L or Neuronal), P/Q-type (Purkinje and Granular), and R-type (dihydropyridine- Resistant) channels display longer-lasting conductance and are expressed throughout the nervous system; T-type (Transient) channels produce short-term conductance and are found in neurons and cells that have pacemaker activity [6]. The activation of the channels results in a Ca^2+^ cascade, which is associated with numerous host physiological functions including excitation–contraction–relaxation coupling of muscles, synaptic transmission, immunoprotection, etc.

Since 1984, it has been known that verapamil, the blocker of VGCCs, inhibits influenza A virus (IAV) infection [7]. Moreover, IAV infection induces Ca^2+^ influx, and the elevated intracellular Ca^2+^ promotes endocytic uptake of IAV [8]. Until recently, the underlying mechanism was revealed by Fujioka et al. [9]. They reported that several VGCC blockers and siRNA against the Ca_V_1.2 channel (L-type channel) inhibited H1N1 and H3N2 IAV infection in multiple cell lines. IAV attaches to the target cells by the viral hemagglutinin protein (HA), which is a type I transmembrane protein embedded in the viral envelope, binding to sialic acids [10]. Fujioka et al. showed that the virus HA binds to domain IV of Ca_V_1.2, which contains two potential sialylated asparagine residues (N1436 and N1487) [9,11]. When Ca_V_1.2 mutants in one or both these residues were replaced with glutamine (N1436Q, N1487Q, and N1436Q + N1487Q), the interaction between HA and the Cav1.2 was attenuated compared with the wild-type fragment. They demonstrated that a VGCC blocker, diltiazem, significantly prolonged the survival of IAV-infected mice and allowed the recovery of the survivors. Therefore, Ca_V_1.2 may serve as a host cell surface receptor that binds IAV and is critical for IAV entry (Figure 2A). New world hemorrhagic fever arenaviruses (NWA) are another virus that is reportedly sensitive to the VGCC blocker club member [12,13]. An siRNA screen with Junín virus glycoprotein-pseudotyped viruses identified that VGCCs are involved in NWA entry. Treatment with the channel blocker gabapentin, an FDA (U.S. Food and Drug Administration)-approved analgesic that targets the channel α2δ2 subunit, protects against NWA infection [13]. This research work demonstrated that the interaction between a virus and VGCCs promotes virus entry at the virus-cell fusion step. 

The effect on VGCCs is not only restricted to virus entry. A high-throughput screening of an FDA-approved drug library for inhibitors of Japanese encephalitis virus (JEV) identified five hit drugs, three of which are VGCCs blockers (manidipine, cilnidipine, and benidipine hydrochloride) [14]. Recombinant viral particles (RVPs) with the luciferase-reporting replicon were used to quantify the efficiency of JEV replication, which confirmed that these drugs inhibited JEV infection at the stage of replication. These drugs were subsequently validated for their antiviral activities against other flavivirus, such as Zika virus (ZIKV), dengue virus (DENV), and West Nile virus (WNV). Similarly, another research group screened the FDA-approved drug library and found that nifedipine and benidipine hydrochloride inhibited severe fever with thrombocytopenia syndrome virus (SFTSV) replication in vitro [12]. Moreover, the retrospective clinical investigation on SFTS patients showed that nifedipine can significantly inhibit SFTSV infection. These studies indicate that the VGCCs blockers are excellent candidates for broad-spectrum anti-virus treatment. Most of these FDA-approved VGCC blockers are clinically used to treat cardiovascular diseases. Similar to a double-edged sword, the cardiovascular effect of VGCC blockers may limit their antiviral application. Therefore, repurposing these drugs requires more analysis before clinical trials. 

Further examples of virus-regulating VGCCs to service their replication can be found in HIV-1 and herpes simplex virus (HSV)-1. Two HIV-1 proteins, the membrane glycoprotein gp120 [15,16] and the transcriptical transactivator Tat [17,18], have been identified to induce the elevation of host intracellular Ca^2+^ in various cell types, including neuronal, immune, and epithelial cells, via targeting the activity of VGCCs. The detailed information about the dysregulation of the L-type calcium channel by Tat can be found in the review [16,17]. 

Taking together, the infections of viruses could increase the host’s intracellular calcium to facilitate viral entry and replication by manipulating host VGCCs. In addition, herpes simplex virus (HSV)-1 could downregulate the VGCCs on infected neuronal cells to escape detection by host cells. T-type Ca^2+^ channels were reported as the targets of HSV-1 in sensory-like ND7-23 cells [19,20]. HSV-1 infection of differentiated ND7-23 cells causes a significant loss of T-type Ca^2+^ channels from the membrane, which depends on viral replication and protein synthesis. This downregulation of T-type Ca^2+^ channel expression may alter the ability of sensory neurons to transmit pain information. Thus, the lower expression of VGCCs may diminish the detection for viral infection by the host, which benefits virus survival and further infection.

Ebola virus (EBOV) used to be considered a VGCC blocker-sensitive virus, and several research groups independently reported that compounds blocking L-type channels (such as verapamil, nimodipine, and diltiazem) inhibited EBOV infection in vivo [21,22,23]. However, gabapentin, representing a fifth distinct class of L-type channel inhibitor, had no effect even at high concentrations. It has been shown that verapamil, nimodipine, and diltiazem also inhibit the activity of two-pore channels (TPCs) [24]. TPCs are intracellular voltage-gated and receptor-operated calcium permeable channels, playing an integral role in membrane trafficking pathways [25,26]. Mouse embryonic fibroblasts (MEFs) lacking TPC1 or TPC2 expression (Tpcn1^−/−^, Tpcn2^−/−^) resisted EBOV infection [27]. It turns out that the target of EBOV may not be classical L-type calcium channels but rather endosomal calcium channels termed TPCs (Figure 2A). The calcium channels inhibitors prevented virus–endosome membrane fusion and virus capsid releasing into the cell cytoplasm, which is a late entry step. EBOV acts on TPCs, which control the movement of endosomes containing virus particles, and thereby facilitate its intracellular trafficking [27,28]. The channel inhibitor, tetrandrine, significantly enhanced the survival of mice challenged with mouse-adapted EBOV without any detectable side effects. This indicates that tetrandrine is highly effective against EBOV disease in mice.

## 4. Store-Operated Calcium (SOC) Channel in Viral Assembly and Egress

SOC channels are the major Ca^2+^ entry pathways in non-excitable cells. The protein Orai1 on the plasma membrane and STIM1 (stromal interaction molecule) on ER are the molecular identities that are responsible for SOC channels activation. The depletion of ER Ca^2+^ stores promotes STIM1 proteins aggregation and interaction with Orai1 to open the channel, mediating Ca^2+^ entry [29,30]. 

Most enveloped viruses are released extracellularly via exocytosis, also called budding, as an analogy of buds in plants [31,32]. The budding process of the enveloped viruses is triggered by a peptide motif (termed late (L) domains), which was discovered in the Gag polyproteins of retroviruses and M (matrix) proteins of rhabdoviruses [33]. These L domains interact with cellular proteins to promote the formation of virus vesicles that bud away from the cytoplasm [32,34]. Research works established that this essential final step of related viruses depends on the host Ca^2+^ signal mediated by SOC channels (STIM1/Orai1).

The research work done on four distinct hemorrhagic fever viruses (Ebolavirus (EBOV), Marburgvirus (MARV), Lassa Virus (LASV), and Junín Virus (JUNV)) demonstrated that EBOV, MARV VP40, and JUNV Z proteins trigger host cell Ca^2+^ signals by activating the ER Ca^2+^ sensing protein STIM1 and the plasma membrane ORAI1 channel. The STIM1/Orai1-mediated Ca^2+^ signal is critical for EBOV and related viruses budding from host cells [35]. The suppression of STIM1 expression and genetic inactivation or the pharmacological blockade of ORAI1 inhibits infections of EBOV, MARV, and JUNV in cultured cells (Figure 2B). Obviously, the matrix proteins or live virus activates STIM1 and the ORAI channel. Similarly to hemorrhagic fever viruses, the HIV-1 matrix protein Gag, directing HIV-1 budding, and mediating VLP (Virus-like Particle) formation also exhibit dependence on Ca^2+^ regulation [36,37]. Therefore, further study is needed to validate the role for STIM1 and the ORAI channel in HIV- 1 budding.

Infections of other enveloped RNA viruses that buds in similar mechanisms may also be inhibited by STIM1 and ORAI1 inhibitors. Indeed, SOC channel antagonists significantly reduced DENV yield [38]. When the human hepatic HepG2 and Huh-7 cells are infected by DENV, STIM1 and ORAI1 were shown to be co-localized in infected cells, indicating activation of the SOC channel [38,39]. Therefore, DENV infection alters cell Ca^2+^ homeostasis possibly via promoting the interaction between STIM1 and ORAI1.

It is possible that the viral proteins trigger the Ca^2+^ depletion in ER or prevent ER refilled with Ca^2+^ to maintain resting ER Ca^2+^ levels [40,41,42]. Alternately, these viral proteins may directly modify STIM1 and uncouple the activation of STIM1 from Ca^2+^ levels [35,43]. Thus, STIM1/ORAI might represent a conserved target to regulating the budding of enveloped RNA viruses and possibly DNA viruses that rely on similar host cellular proteins for budding. The example is that the hepatitis B virus X (HBx) protein upregulates the activity of the STIM1–ORAI1 channel complex [44]. The mechanisms by which these viruses activate STIM1 and ORAI1 represent novel therapeutic targets for controlling budding.

Calcium influx through the SOC channel also contributes to the elevated cytosolic calcium concentration induced by rotavirus (RV) infection. It has been well established that the dramatically increased cellular Ca^2+^ is the hallmark of rotavirus (RV) infection [45,46]. Rotavirus nonstructural protein 4 (NSP4) is an ER-localized viroporin that functionally depletes ER calcium. Thus, in rotavirus-infected cells, STIM1 is constitutively active and colocalizes with the ORAI1 channel. The knockdown of STIM1 or the pharmacological inhibition of SOC channels significantly reduced rotavirus yield, indicating that the SOC channel plays a critical role in the RV lifecycle [47,48].

## 5. Host Transient Receptor Potential (TRP) Channels and Receptor-Operated Calcium (ROC) Channels 

### 5.1. TRP Channels

The TRP channel is a non-selective cation channel predominately permeable for Ca^2+^ [49,50]. It is divided into six sub-families according to their amino acid sequence: TRP canonical or classical (TRPC), TRP vanilloid (TRPV), TRP melastatin (TRPM), TRP ankyrin (TRPA), TRP polycystin (TRPP), and TRP mucolipin (TRPML) [50]. They are ubiquitously expressed in different tissues and cell types and are a key player in the regulation of intracellular calcium. Nevertheless, reports about virus control TRP channels are not numerous.

TRPV4 mediates intracellular Ca^2+^ signals in response to several stimuli, including hypotonic cell swelling, mechanical forces, moderating heat, and UVB radiation [51]. When cells are exposed to the Zika virus or the purified viral envelope protein, TRPV4 mediates Ca^2+^ influx and drives the nuclear translocation of DEAD-box RNA helicase (DDX3X) [52]. DDX3X is an ATP-dependent RNA helicase from the DEAD-box helicase family and is involved in multiple stages of the RNA metabolism, from transcription to translation [53,54]. Moreover, diverse RNA viruses hijack DDX3X to multiply efficiently. Targeting TRPV4 reduces the infectivity of dengue, hepatitis C, and Zika viruses [52]. Overall, the research work demonstrated the role of TRPV4 in the regulation of DDX3X-dependent control of the RNA metabolism and viral infectivity [52].

The TRPV1, TRPA1, and TRPM8 channels are directly activated by chemical, thermal, and mechanical stimuli. They are potentially associated with respiratory virus-induced airway hypersensitivity [55]. The infection of respiratory viruses, such as respiratory syncytial virus (RSV), measles virus (MV), and human rhinovirus (HRV), was reported to increase the expression of these channels in sensory neurons and human bronchial epithelium cells [56,57]. The increase in TRPA1 and TRPV1 levels can be mediated by soluble factors induced by infection, whereas TRPM8 requires virus replication [56,57]. These reports may explain the possible mechanism by which respiratory viruses induce cough. Alternatively, the up-regulated TRP channels may be utilized by the respiratory viruses to create a favorable calcium environment. Further investigation is required to determine the possible pathway by which this happens.

### 5.2. N-Methyl-D-Aspartate (NMDA) Receptors 

NMDArs and IP_3_Rs are the two receptor-operated calcium (ROC) channels reported to be involved in virus–host interactions. NMDAr, which is activated by the endogenously synthesized excitant amino acid, glutamate, is widely expressed throughout the mammalian central nervous system and is particularly enriched in the cerebral cortex and hippocampus.

The Zika virus infection is a “neurodegenerative” disease. Costa and colleagues showed that blockage of the NMDAr channel activity with FDA-approved memantine or other antagonists prevents neuronal cell death and microgliosis induced by Zika in vitro and in vivo, without affecting the ability of Zika to replicate in the host [58,59]. It seems that NMDAr mainly contributes to Zika, triggering the neuronal cell death progress. Anyway, the blockade of NMDAr may be a viable treatment for patients at risk for Zika infection-induced neurodegeneration [59]. NMDAr blockade also significantly abrogated neuronal cell death and inflammatory response triggered by JEV infection [60]. Therefore, NMDAr are probably common attack targets of flavivirus, inducing host neuron cell death. 

### 5.3. IP_3_ Receptors

The engagement of receptors or agonist binding to the cell surface receptors activate phospholipase C (PLC), which hydrolyses phosphatidylinositol-4,5-biphosphate (PIP_2_) to produce IP_3_. The activation of IP_3_R triggers Ca^2+^ release from stores and further increases IP_3_R’s sensitivity to Ca^2+^. Targeting IP_3_Rs is more universal among different virus types. In turn, this process generates a more profound effect on host cell physiology, including metabolic stress, neurotoxicity, and enterotoxicity. Modifying IP_3_ and directly interfering with IP_3_R are the main ways in which viruses affect calcium release through IP_3_R. For example, human cytomegalovirus (HCMV) UL37x1 protein interacts with the host P2Y2 purinergic receptor to increase intracellular Ca^2+^ levels via the PLC–IP_3_ pathway, and this activity is important to viral replication [61]. HIV-1 Gp120 and Tat upregulate intracellular IP_3_ [37,62], while HIV Nef [63,64] and p12(I) human T-cell lymphotropic virus type 1 (HTLV-1) [65] activate IP_3_R directly as an agonist. Glycoprotein E of HSV redistributes the density of IP_3_Rs within infected cells [66]. Some virus proteins activate IP_3_R through both ways. Enterotoxin NSP4 of RV and HCMV UL37x1-encoded protein increases the basal permeability of the ER as IP_3_ diffuses and binds to IP_3_Rs to stimulate Ca^2+^ release. These virus proteins deplete ER Ca^2+^ storage during early stages of viral infection to increase the replication ability of viruses. On the other hand, the depletion of ER calcium storage triggers the calcium influx through SOCs (STIM1/ORAI1).

## 6. Calcium Pumps

The calcium pumps (Ca^2+^ ATPase) and the Na^+^–Ca^2+^ exchanger are the main regulators of intracellular Ca^2+^ concentrations [67]. Three Ca^2+^ ATPase types (pumps) have been described in animal cells located in the membranes of the sarcoplasmic reticulum (SERCA pump), in those of the Golgi network (Secretory Pathway Ca^2+^ ATPase, SPCA pump), and in the plasma membrane (PMCA pump) [68]. At the end of a stimulus, these pumps participate in returning the cell to its resting state by the decrease in the cytosolic Ca^2+^ concentration. For example, SERCA transports Ca^2+^ from the cytoplasm into the lumen of the ER. PMCA pumps function to remove Ca^2+^ from the cell. They serve as the main regulators of intracellular and extracellular Ca^2+^ concentrations. All three calcium pumps are reported to be involved in different viral lifecycles.

Maturation is the last assembly step of a virus particle. In this step, the virus acquires the infectivity, which is essential for its lifecycle. The results obtained from a genome-wide knockout screen indicated that SPCA1 in the Golgi network is critical for human respiratory syncytial virus (RSV) infection [69,70,71]. Further study demonstrated that similar to RSV, other viruses from the Paramyxoviridae, Flaviviridae, and Togaviridae families also failed to spread in SPCA1-deficient cells. Studies on this mechanism revealed that Ca^2+^ pumped into Golgi by SPCA1 is the trigger to produce normal functional viral glycoproteins that are essential for virus spread. Therefore, SPCA1 may be a promising target for therapeutic intervention against a diverse set of viruses (Figure 2B). However, in this study, herpes simplex 1 (HSV-1), vesicular stomatitis (VSV), bunyamwera (BUNV), and LCMV were unaffected by the loss of SPCA1. That is probably because the maturation of these viral glycoproteins requires different triggers.

Cui et al. [72] screened the natural product library and found that cyclopiazonic acid, an inhibitor of SERCA, was shown to have activity in the low micromolar range against RSV strains at the step of virus genome replication. Further study found that another SERCA inhibitor BHQ had a similar effect [72]. It is probable that SERCA inhibitors prevent cytosolic calcium returning to the ER from the cytosol via SERCA, resulting in an increase in intracellular calcium concentration. A continuous higher concentration of intracellular calcium may impair viral genome replication and/or transcription, thereby reducing virus yield.

## 7. Conclusions

The journey of viral infections is also a procedure during which a virus grabs the host intracellular calcium signaling system and homeostasis to benefit its own lifecycle. The disturbance of the host Ca^2+^ system by virus may suppress T-cell responsiveness, antiapoptotic, and other protentional functions. In this review, we summarized the calcium permeable channels and calcium pumps located on host cell membranes that may be potential therapeutic targets for viral infections (Table 1). However, the calcium channel family is continuously being refreshed with the discovery of new members, and the effect of viruses on the host intracellular calcium environment is not often achieved through a single pathway; it is more complicated. Both upstream messengers and downstream Ca^2+^ binding proteins are targets for pharmacological intervention. Thus, explicating the entire picture of all aspects of the calcium–virus interplay at the molecular level will be a challenge in the future.

The prevention of and therapeutic efforts against virus infections are often challenged by the high mutation rates of viral proteins. However, those host calcium apparatus proteins required by viral replication and transmission are highly conserved, essentially immutable, and thus are potentially an Achilles’ heel of virus infection. As described above, almost all the calcium permeable channels or calcium pumps are possible attack targets of viruses and customized host intracellular calcium signaling via effects on these channel proteins. Targeting those host cellular proteins required by virus infection may be one treatment strategy with therapeutic potential, which may avoid or delay the development of resistance. The identified calcium-permeable proteins discussed above offer suitable target proteins for inhibiting virus infection. VGCCs and NMDA receptors are two of most well studied targets because of the availability of specific channel antagonists. Moreover, some of those compounds are FDA-approved drugs with clinical application. For example, verapamil, amlodipine, verapamil, and diltiazem, which are widely used to treat cardiovascular disease, have shown inhibition for various virus infection. Memantine, which is an FDA-approved Alzheimer’s drug, reduces neurological complications associated with Zika virus infection. Obviously, this direction depends on the identification of selective and potent small molecule modulators of calcium channels. Currently, aside from some blockers for VGCCs and the NMDA receptor, almost no other channel modulators can be used in clinics, although important progress has been made in recent years toward high quality and widely available channel modulators. Furthermore, modulators should be generated with properties that make them suitable for in vivo use. This review outlines promising progress made with the potentiation for a host cell calcium channels modulator to become an antiviral drug. We believe that modulators for host calcium channels are the pots of gold for antiviral drug development.

## Figures and Tables

**Figure 1 cells-09-00094-f001:**
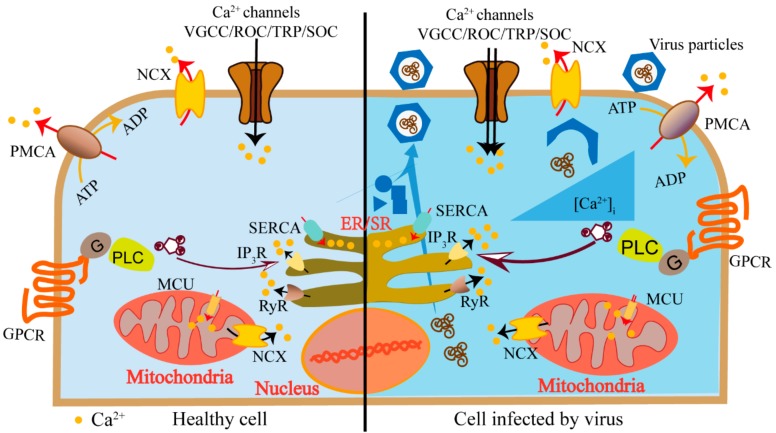
Schematics of host cell elevated cytosolic calcium concentration induced by a virus. Calcium channels (voltage-gated calcium channels (VGCCs), receptor-operated channels (ROC), store-operated Ca^2+^ (SOC), channels and transient receptor potential (TRP) channels) mediate the entry of Ca^2+^ from extracellular medium (black arrows). The IP_3_ receptor (IP_3_R) and the ryanodine receptors (RyR) on the endoplasmic reticulum (ER) mediate the release of Ca^2+^ from internal stores (black arrows). Calcium pumps (the plasma membrane Ca^2+^-ATPase (PMCA), sarcoplasmic/endoplasmic reticulum Ca^2+^-ATPase (SERCA)) and the Na^+^/Ca^2+^ exchanger (NCX) are responsible for transporting Ca^2+^ from the cytosol to external medium or into cellular calcium stores (red arrows). Viruses utilize these calcium components to elevate cytosolic calcium concentration to activate Ca^2+^-dependent/sensitive enzymes and transcriptional factors to promote virus replication (right panel).

**Figure 2 cells-09-00094-f002:**
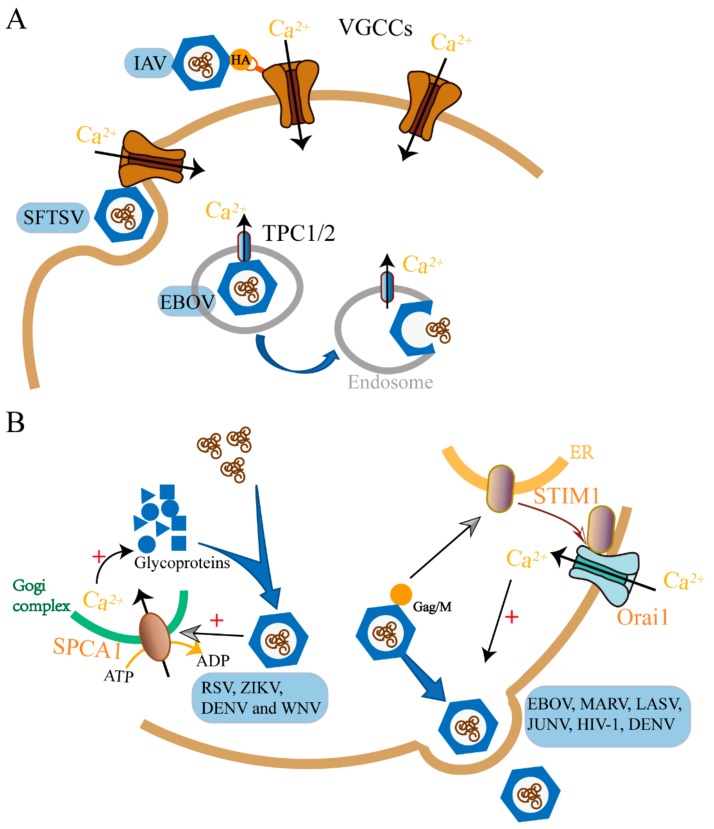
Examples of viruses interplaying with host calcium channels or pumps to achieve viral entry (**A**) and release (**B**). VGCCs are important for influenza A virus (IAV) and severe fever with thrombocytopenia syndrome virus (SFTSV) entry into the host cell as well as TPC1/2 for EBOV (**A**). RSV, Zika virus (ZIKA), dengue virus (DENV) and West Nile virus (WNV) hijack SPCA1 to facilitate their release as well as Ebola virus (EBOV), MARV, LASV, JUNV, HIV-1 and DENV manipulates STIM1/ORAI1 (**B**). For a complete list of definitions, see Table 1.

**Table 1 cells-09-00094-t001:** Calcium channels/pumps utilized by a virus.

Cellular Targets	Virus	Consequences [Ref.]
VGCCs	IAV	Ca_V_1.2 serves as a host cell surface receptor that binds IAV and is critical for IAV entry [9].
	SFTSV	Benidipine hydrochloride, VGCC blocker, inhibits SFTSV infection via impairing virus internalization and genome replication [12].
	NWV	Virus binds to VGCCs and promotes virus entry at the virus–cell fusion step [13].
	Flavivirus (JEV, ZIKV, DENV, and WNV	VGCCs blockers inhibit flavivirus (JEV, ZIKV, DENV and WNV) infection at the stage of replication [14].
	HIV-1	Tat/gp120 overactivate VGCCs [15,16,17].
	HSV-1	HSV-1 downregulates the Ca_V_3.2 channel and diminishes the detection of viral infection by host [19,20].
TPCs	EBOV	Facilitates virus–endosome membrane fusion and releases of virus capsid into the cell cytoplasm [27].
STIM1/ORAI1	EBOV, MARV, LASV, JUNV, HIV-1, DENV, and HBV	Promote virion assembly and budding [35,36,37,38,44].
TRPV4	ZIKV	Activation of TRPV4- releases DDX3X and promote the viral RNA metabolism [52].
NMDAr	ZIKV, JEV	NMDAr contributes to ZIKA by triggering the neuronal cell death progress [58,60].
	HIV-1	Increases Ca^2+^ influx [62].
IP_3_R	HIV-1, HSV, HRV, and HCMV	These viral proteins deplete ER Ca^2+^ store during early stages of viral infection to increase the replication ability of viruses [37,62,63,64,66].
SPCA1	RSV, ZIKV, DENV and WNV	Trigger to produce functional viral glycoproteins that are essential for virus spread [70]

Abbreviations: Influenza A virus (IAV), Severe fever with thrombocytopenia syndrome virus (SFTSV), New world hemorrhagic fever arenaviruses (NWV), Japanese encephalitis virus (JEV), Zika virus (ZIKV), Dengue virus (DENV), West Nile virus (WNV), Herpes simplex virus (HSV)-1, Ebolavirus (EBOV), Marburgvirus (MARV), Lassa Virus (LASV), Junín Virus (JUNV), Hepatitis B virus (HBV), Respiratory syncytial virus (RSV), Human rhinovirus (HRV), and Human cytomegalovirus (HCMV). Voltage-gated calcium channels (VGCCs), Two-pore channels (TPCs), Stromal interaction molecule 1 (STIM1), Transient receptor potential vanilloid 4 (TRPV4), N-methyl-D-aspartate receptor (NMDAR), IP_3_ receptor (IP_3_R), Secretory pathway Ca^2+^-ATPase (SPCA1). DEAD-box RNA helicase (DDX3X), Endoplasmic reticulum (ER).

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
