# Peer review of "Host Calcium Channels and Pumps in Viral Infections"

_cells, 2019, doi:10.3390/cells9010094_

Round 1

Reviewer 1 Report

The manuscript by Chen et al. reviews the present knowledge on the interaction between viral infection and the cellular Ca2+ signalling machinery. The subject is adequate and the literature is well reviewed. However, reading becomes quite difficult because of the poor English grammar throughout the manuscript, which should be fully corrected before publication.

In addition, there are several points that should be improved or corrected:

Lines 56-58. “The IP3 receptor (IP3R) and the ryanodine receptors (RyR) on ER mediate the release of Ca2+ from internal stores, which are trigged by intracellular messengers IP3 and lower concentration of Ca2+ in the cytosol.” I don’t understand what it means this “lower concentration of Ca2+ in the cytosol”. IP3R are activated by IP3, and both IP3R and RyR are activated by an increase of cytosolic Ca2+, not by a lower concentration of Ca2+ in the cytosol. Figure 1. The word Mirochiondria should be changed by Mitochondria. In addition, the MCU channel seems to be placed in the outer mitochondrial membrane, when both MCU and the mitochondrial NCX are in the inner one. I understand that there is no space, but MCU should at least end in the mitochondrial matrix. Figure 2. The figure is divided in A and B, saying that A includes channels or pumps producing Ca2+ entry, and B those that produce Ca2+ release. However, there are systems producing Ca2+ entry and Ca2+ release in both A and B. That division is wrong. In addition, a complete description of the figure, describing all the elements, appears necessary. Lines 133-134. When it is stated that VGCC blockers are excellent candidates for broad-spectrum anti-virus treatment, it should be taken into account that these drugs produce important effects on blood pressure and cardiac function, and therefore their use to treat viral infections has important limitations. In the paragraph starting at line 193, several statements are made without providing references for them. In line 248-9 it says “Specific agonist binding to cell surface G protein coupled receptor (GPCR) activate phospholipase C-γ (PLC-γ)”. This is not correct. GPCR agonist may only activate PLC-beta. PLC-gamma is activated by tyrosine kinase receptors. In line 261, it says “ROCs (STIM1/Orai1)”. It should be SOCs, not ROCs. Lines 283-286. It says: “Cui R et al. [63] screened natural product library and found that cyclopiazonic acid, an inhibitor of calcium-dependent ATPases, was shown to have activity in the low micromolar range against RSV strains belonging to both A and B subgroups and PIV-3[66]. This might be explained by the above revealed mechanism: the inhibition of SPCA1 prevent the production of viral glycoproteins”.
Cyclopiazonic acid is a specific inhibitor of SERCA, not of calcium-dependent ATPases in general. In particular, it does not inhibit SPCA1 and therefore inhibition of SPCA1 cannot explain its effects. The whole paragraph is quite confusing because after that it talks about specific inhibitors of SERCA, but without referring to cyclopiazonic acid. It should be rewritten.

Author Response

Response to Reviewer 1 Comments

Point 1: The manuscript by Chen et al. reviews the present knowledge on the interaction between viral infection and the cellular Ca2+ signalling machinery. The subject is adequate and the literature is well reviewed. However, reading becomes quite difficult because of the poor English grammar throughout the manuscript, which should be fully corrected before publication.

Response 1: Thank you for your positive evaluation of our review manuscript. We used the English editing service provided by MDPI to improve the language as requested.

Point 2: Lines 56-58. “The IP3 receptor (IP3R) and the ryanodine receptors (RyR) on ER mediate the release of Ca2+ from internal stores, which are trigged by intracellular messengers IP3 and lower concentration of Ca2+ in the cytosol.” I don’t understand what it means this “lower concentration of Ca2+ in the cytosol”. IP3R are activated by IP3, and both IP3R and RyR are activated by an increase of cytosolic Ca2+, not by a lower concentration of Ca2+ in the cytosol.

Response 2: Thank you for pointing out it. We revised the MS and corrected this sentence (lines 59-61)

Point 3: “Figure 1. The word Mirochiondria should be changed by Mitochondria. In addition, the MCU channel seems to be placed in the outer mitochondrial membrane, when both MCU and the mitochondrial NCX are in the inner one. I understand that there is no space, but MCU should at least end in the mitochondrial matrix.”

Response 3: We corrected the misspelling of “mitochondria” in figure 1. The MCU was placed in the inner mitochondrial membrane in figure 1 as requested.

Point 4: Figure 2. The figure is divided in A and B, saying that A includes channels or pumps producing Ca2+ entry, and B those that produce Ca2+ release. However, there are systems producing Ca2+ entry and Ca2+ release in both A and B. That division is wrong. In addition, a complete description of the figure, describing all the elements, appears necessary.

Response 4: We made Figure 2 to illustrate examples of viruses interplaying with host calcium channels or pumps to achieve virus entry (A) and release (B), rather than the Ca2+ entry and release. We added a complete description for figure 1 as requested (lines126-128).

Point 5: Lines 133-134. When it is stated that VGCC blockers are excellent candidates for broad-spectrum anti-virus treatment, it should be taken into account that these drugs produce important effects on blood pressure and cardiac function, and therefore their use to treat viral infections has important limitations.

Response 5: We agree with reviewer that the cardiovascular effects may limit the application of VGCC blockers. We revised the manuscript and added sentences to state this point now (lines141-143).

Point 6: In the paragraph starting at line 193, several statements are made without providing references for them.

Response 6: We revised the manuscript and added missing references as requested (line 204, ref. [40-42] and line 205, ref. [35, 43]).

Point 7: In line 248-9 it says “Specific agonist binding to cell surface G protein coupled receptor (GPCR) activate phospholipase C-γ (PLC-γ)”. This is not correct. GPCR agonist may only activate PLC-beta. PLC-gamma is activated by tyrosine kinase receptors.

Response 7: We rephrased this sentence and corrected the mistake (lines 260- 263).

Point 8: In line 261, it says “ROCs (STIM1/Orai1)”. It should be SOCs, not ROCs.

Response 8: Thank you for pointing out the mistake. We corrected this mistake (line 276).

Point 9: Lines 283-286. It says: “Cui R et al. [63] screened natural product library and found that cyclopiazonic acid, an inhibitor of calcium-dependent ATPases, was shown to have activity in the low micromolar range against RSV strains belonging to both A and B subgroups and PIV-3[66]. This might be explained by the above revealed mechanism: the inhibition of SPCA1 prevent the production of viral glycoproteins”. Cyclopiazonic acid is a specific inhibitor of SERCA, not of calcium-dependent ATPases in general. In particular, it does not inhibit SPCA1 and therefore inhibition of SPCA1 cannot explain its effects. The whole paragraph is quite confusing because after that it talks about specific inhibitors of SERCA, but without referring to cyclopiazonic acid. It should be rewritten..

Response 9: We rewrote this paragraph as suggested by reviewer (lines 298-304).

Reviewer 2 Report

The review of Chen et al deals with the impact of calcium homeostasis and channels in viral infections. In general, this is a very interesting and hot topic which – in my opinion - perfectly fits in the special issue.  

The review has a clear structure, it summarizes the most important molecular mechanisms and gives a good overview of the recent literature.  It is interesting for a board readership, especially for scientists dealing with calcium homeostasis and viral infections. I would like to point out that the authors tried to elucidate the different molecular mechanisms and illustrate these findings in 2 nice and clear figures.

I have only very minor suggestions before this review is suitable for publication

Style and language are mostly fine. However, at some points, the reader might realize that the authors are not native speakers. I would recommend that a native speaker go through the manuscript and amend and “smooth” a little bit the wording.

The authors divide the review by the ion channels, which are affected, which is a good approach to get this topic structured. I recommend adding a short table or conclusion at the end of each paragraph where the main literature is listed.

Probably I have overseen it, but I would recommend adding the following papers: Rotavirus Calcium Dysregulation Manifests as Dynamic Calcium Signaling in the Cytoplasm and Endoplasmic Reticulum by Alexandra L. Chang-Graham Scientific reports 2019 P2Y2 purinergic receptor modulates virus yield, calcium homeostasis, and cell motility in human cytomegalovirus-infected cells by Saisai Chen, PNAS 2019

Author Response

Response to Reviewer 2 Comments

Point 1: The review of Chen et al deals with the impact of calcium homeostasis and channels in viral infections. In general, this is a very interesting and hot topic which – in my opinion - perfectly fits in the special issue. The review has a clear structure, it summarizes the most important molecular mechanisms and gives a good overview of the recent literature. It is interesting for a board readership, especially for scientists dealing with calcium homeostasis and viral infections. I would like to point out that the authors tried to elucidate the different molecular mechanisms and illustrate these findings in 2 nice and clear figures. I have only very minor suggestions before this review is suitable for publication. Style and language are mostly fine. However, at some points, the reader might realize that the authors are not native speakers. I would recommend that a native speaker go through the manuscript and amend and “smooth” a little bit the wording.

 Response 1: Thank you for your positive evaluation of our review manuscript. We are not native speakers indeed. We used “English editing service” provided by MDPI to improve the language as required.

Point 2: The authors divide the review by the ion channels, which are affected, which is a good approach to get this topic structured. I recommend adding a short table or conclusion at the end of each paragraph where the main literature is listed.

Response 2: Thanks for your comment. We had a summary table (Table 1. Calcium channels/pumps utilized by virus) in the “Conclusions” part, including main involved calcium channels with the literatures (lines 317-324). As such, we did not repeat the information in each paragraph.

Point 3: Probably I have overseen it, but I would recommend adding the following papers: Rotavirus Calcium Dysregulation Manifests as Dynamic Calcium Signaling in the Cytoplasm and Endoplasmic Reticulum by Alexandra L. Chang-Graham Scientific reports 2019 P2Y2 purinergic receptor modulates virus yield, calcium homeostasis, and cell motility in human cytomegalovirus-infected cells by Saisai Chen, PNAS 2019

Response 3: We thank the reviewer to supply the most recent papers on this topic and added these two papers (line 217, ref. [48] and 267, ref. [61]) as suggested.